# Path Planning with Time Windows for Multiple UAVs Based on Gray Wolf Algorithm

**DOI:** 10.3390/biomimetics7040225

**Published:** 2022-12-03

**Authors:** Changchun Zhang, Yifan Liu, Chunhe Hu

**Affiliations:** 1School of Technology, Beijing Forestry University, Beijing 100083, China; 2Research Center for Biodiversity Intelligent Monitoring, Beijing Forestry University, Beijing 100083, China

**Keywords:** multi-UAV, path planning, time windows, swarm intelligence, GWO

## Abstract

The Gray Wolf (GWO) algorithm aims to address the path planning problem of multiple UAVs, and the scene setting is mainly to avoid threats, meet the constraints of UAVs themselves and avoid obstacles between UAVs. The scene setting is relatively simple. To address such problems, the problem of time windows is considered in this paper, so that the UAV can arrive at the same time, and the Gray Wolf algorithm is used to optimize the problem. Finally, the experimental results verify that the proposed method can plan a safe flight path in the process of multi-UAV flight and reach the goal point at the same time. The mean error of flight time between UAVs of the GWO is 0.213, which is superior to PSO (0.382), AFO (0.315) and GA (0.825).

## 1. Introduction

Unmanned aerial vehicles (UAVs) are a crucial symbol of military strength of all countries in the world, which can strengthen nation defense construction and accelerate information construction. The development of UAVs is driven by multi-disciplines and high technology in the information age. As UAVs are popularized in various fields—such as observations of Water Surface Elevation and Bathymetry and Shallow Water Bathymetry and Object Detection, and so on [1,2,3,4,5]—the working environment of a UAV becomes more and more complex, which is difficult to be solved by a single UAV. Therefore, it is more helpful to solve the increasingly complex mission environment and mission requirements for multiple UAVs to perform flight missions together. In the multi-UAV system, each UAV usually has no access to global information. By exchanging the information of neighbors, the group behavior of the cluster and the global coordinated goals can be realized [6].

When multiple UAVs perform tasks, they need to plan the path information. After planning the path information, they need to exchange information with neighbors and make certain adjustments according to their information, so as to better complete the task. Multi-UAV path planning requires the generation of flight paths from the starting position to the target position for each UAV, and requires that the total path cost of the cluster route is the lowest (lower), which can realize the mutual avoidance of UAV in the cluster and avoid collision with the environment.

Multi-UAV path planning belongs to the multi-UAV autonomous decision making and planning problem, which is essentially an optimization problem, and can be solved by using the optimization strategy. In recent years, many optimization algorithms have been proposed to solve such problems; for example, the Dijkstra algorithm, A* algorithm and artificial potential field method in the field of classical search [7,8,9]. However, one of the most important characteristics of multiple UAVs is to realize the group behavior of the UAV group with the help of the local interaction between UAVs, which has distributed characteristics, while classical search algorithms do not have distributed characteristics. To solve this problem—inspired by the fact that in nature, and in order to make up for the limited ability of individuals, many biological populations can present some kind of group behavior through the communication and cooperation between individuals or local regions—scholars at home and abroad have proposed a series of swarm intelligence methods, such as the Particle Swarm Optimization, PSO algorithm [10], Ant Colony Optimization, ACO algorithm [11,12,13], Artificial fish swarm, AFO algorithm [14,15,16], pigeon-inspired optimization, PIO algorithm [17], Firefly algorithm, FA algorithm [18], Genetic Algorithm and GA algorithm, and so on [19]. Coyotes have become the masters of the prairies, through their own strong cognitive ability and tight organization structure within the team. Under the severe living environment in nature, coyotes have created amazing cooperative hunting methods. Inspired by wolf hunting behavior, Mirjalili, A. and other scholars proposed a swarm intelligence algorithm based on Gray Wolf optimization, the GWO algorithm [20]. Since the Gray Wolf algorithm came out, it has attracted extensive attention of many scholars due to its good performance. The Gray Wolf algorithm has the characteristics of simple structure, fewer parameters to be adjusted and being easy to implement. Moreover, due to the adaptive convergence factor and information feedback mechanism in the algorithm design of the Gray Wolf algorithm, it can achieve a balance between local optimization and global search, and to a certain extent, can solve the common problem of the swarm intelligence optimization algorithm trapped in local optimization. In terms of function optimization, relevant studies have proven that the convergence speed and solution accuracy of GWO are superior to PSO, AFO and GA. After much research, the swarm intelligence algorithm is considered the most suitable method for UAV cluster path planning, due to its natural advantages of being distributed, self-organizing and scalable.

Currently, the GWO algorithm is mainly applied to multiple UAV route planning problems under a simple scenario, while for complex scenarios using GWO, solving the problem of UAV route planning research does not take much. To address this problem, the article sets UAVs to meet at the same time from different starting points and target points, at the same time, for each drone planning flight path. In addition, a series of performance constraints of UAVs are satisfied, and the GWO algorithm is used to plan the flight path with a faster convergence speed and lower flight path cost.

This paper is divided into five chapters. The first chapter is the relevant introduction of the research content; the second chapter uses the mathematical method to describe the problem; chapter 3 mainly introduces the Gray Wolf algorithm; chapter four analyzes the results; and chapter 5 is a summary and outlook.

## 2. Problem Statement

The chapter introduces graph theory basis, UAV restraint information and environment information and fitness, which enables related researchers to better understand the proposed methods and conclusion.

### 2.1. Graph Theory Basis

The multi-UAV system, composed of multiple UAVs, is a distributed system. The problem of path planning for multiple UAVs is essentially a distributed optimization problem. The distributed optimization problem needs to study the network, in which the topology is usually represented by graphs. This part will make a brief introduction to graph theory, so that readers can better understand the following content.

Graph is usually represented by G={V,ε,A}, *V =* {1, 2, …, *N*} is usually represented union of nodes, ε={(i,j), i,j∈V}⊆V×V is set of edges, and $A={\left[a_{ij}\right]}_{N\times N}$ an adjacency matrix. Here we only discuss digraphs, which can be used to model the mutual communication between nodes, and if *j* is a neighbor of *i*, then the corresponding element is $a_{ij}\neq 0$ of the adjacency matrix (usually set to 1) [21,22].

For any node, the node *i* in-degree is defined as $deg_{out}\left(t\right)=\overset{N}{\underset{j=1}{\sum }}a_{ij}$, and the node *i* in degree is defined as $deg_{in}\left(t\right)=\sum_{j=1}^{N}a_{ij}$. A graph is said to be balanced if the in-degree and out-degree of any node of the graph are equal, as shown in Figure 1. The multi-UAV system studied in this paper is essentially a multi-agent system. There are *N* agents in the network, and *i* represents the $agent_{i}$. In this multi-agent system, agents will exchange information and form a communication network topology, and this network topology is the balance diagram [23,24,25].

### 2.2. UAV Restraint Information and Environment Information

In the problem of UAV flight path planning, the establishment of a UAV planning space environment and a flight path evaluation index is the prerequisite for flight path planning.

**Planning space model.** Different environmental information can directly affect the result of flight path planning and the effect of mission execution, so it is very important to accurately model the planned space environment.

In this paper, the scenario we study is a mountain environment unmanned aerial vehicle (UAV) delivery problem with time windows, in particular: when there is a mountain in the disaster; the need of UAV to deliver aid; a need to be in place to move supplies at a specified time; controlled speed at the same time; the height; angle; and distribution of the weight of the constraint. In order to effectively simulate the mountainous environment when the UAV performs the mission, the terrain mathematical model can be obtained by modeling the obstacles in the mountainous environment [26]
(1)$$z\left(x,y\right)=h_{0}+\sum_{j=1}^{N}h_{j}^{\max}*\exp\left\{-{\left[\frac{k_{j}^{x}\left(x-x_{j}^{\max}\right)}{x_{j}^{\max}}\right]}^{2}-\left[\frac{k_{j}^{y}\left(y-y_{j}^{\max}\right)}{y_{j}^{\max}}\right]\right\}\;\;$$
where $h_{0}$ is benchmark terrain height, N is number of peaks, $h_{j}^{\max}$ is the j peak vertex height, $x_{j}^{\max}$ and $y_{j}^{\max}$ are the j abscissa and ordinate of the peak vertex, and $k_{j}^{x}$ and $k_{j}^{y}$ are the j gradient-related quantities of peaks along the x and y axes.

Maximum turning angle. In the actual flight, the angle of UAV is limited, and can only fly within a certain angle range. Different UAV angles have different constraints.$\;A\left(x_{i-1},y_{i-1},z_{i-1}\right)$ is the previous route, B$\left(x_{i},y_{i},z_{i}\right)$ stands as the current route, and $C\left(x_{i+1},y_{i+1},z_{i+1}\right)$ indicates the next route. Remember $a_{i}={\left[x_{i}-x_{i-1},y_{i}-y_{i-1}\right]}^{T}$, $\varphi_{\max}$ is the maximum allowable turning angle, requiring each turning angle to be subject to the rules which $\varphi_{\max}$ or less. The constraints can be expressed as
(2)$$\cos\left(\varphi \right)=\frac{a_{i}^{T}a_{i+1}}{\left|a_{i}\right|\left|a_{i+1}\right|}\geq \cos\left(\varphi_{\max}\right)\;\left(i=2,\ldots ,n\right)$$
where $\left|a_{i}\right|$ is the length of the vector.

**Maximum angle of climb/angle of dive.**
During flights, ideal flight situations do not exist; UAVs usually have to avoid obstacles by climbing and diving. The maximum climb/dive angle is determined by the UAV’s own ability, which limits the maximum angle at which the final track can climb and dive in the altitude direction (*z*). Assuming that the maximum allowable climb/dive Angle is
θ, the constraint can be expressed as:
(3)$$\frac{\left|z_{i}-z_{i-1}\right|}{\left|a_{i}\right|}\leq \tan\left(\theta \right)\;\left(i=2,\ldots ,n\right)$$

**Minimum turning radius.** Considering the indicator is vital for path planning, as the turning radius is too small to ignore some track points. To prevent the turning radius from destroying path planning, we set a minimum turning radius, which can be calculated by the following formula
(4)$$\varphi \geq 2\arcsin\left(\frac{r_{\min}}{r_{d}+r_{\min}}\right)$$
where $r_{\min}$ is the minimum turning radius of UAV.

**Minimum and maximum flight speed.** When UAVs fly in a complex environment, the flight speed of the UAV can be too quick to avoid obstacles, but too slow to finish the task; thus, the flight speed of the UAV needs to be kept in a limited range. Remember that the flying speed of UAV is v, the maximum flying speed is $v_{\max}$ and the minimum flying speed is $v_{\min}$, and then the constraint can be expressed as:(5)$$v_{\min}\leq v\leq v_{\max}$$

**Minimum flight height.** UAVs belong in the sky—the land isn’t their home. Setting a minimum flight height is necessary. Remember that the flight height is H and the minimum flight height is $H_{\min}$, and then the constraint can be expressed as:(6)$$H\geq H_{\min}$$

**Obstacle avoidance constraint between UAVs.** UAVs must be prevented from colliding with each other. Assuming that each UAV is composed of N track points: the track points of each UAV can be represented by $P_{n}=\left(X_{n},Y_{n},Z_{n},\right)\;\left(n=1,\;2\ldots i\ldots j\right)$; the track points of each UAV in the X direction are denoted as $X_{i}=\left(x_{1},\ldots ,\;x_{i}\;\ldots ,x_{n}\right)$; the track points of each UAV in the Y direction are denoted as $Y_{i}=\left(y_{1},\ldots ,y_{i}\;\ldots ,y_{n}\right);\;\mathrm{and}\;t$he track points of each UAV in the Z direction are denoted as $Z_{i}=\left(z_{1},\ldots ,z_{i}\;\ldots ,z_{n}\right)$, The intersection between the set of trajectory points in each dimension of each UAV is 0, which means that the trajectory points do not coincide. Then, the constraint can be expressed as:(7)$$X_{i}\cap X_{j}=0\;$$
(8)$$Y_{i}\cap \;Y_{j}=0\;$$
(9)$$\;Z_{i}\cap \;Z_{j}=0$$

### 2.3. Fitness of Unmanned Aerial Vehicle

The fitness is a vital index to evaluate the performance of the algorithm. The fitness determines the quality of the optimization result. The optimization objective function of the Gray Wolf algorithm is the fitness function. The fitness of a UAV is determined by both mileage and fuel consumption. The fitness can be expressed as follows:(10)fitness=k×mile+(1−k)×oil 
(11)oil=K×mile
mile represents the mile of UAV, and oil represents the oil of UAV. The value of k ranges from 0 to 1, and the value of k determines whether fitness focuses more on mileage or fuel consumption. K is how much oil is consumed per thousand miles.

The calculation of the mileage depends on the track points in the space; assuming that there are n track points in the space, each track points can be expressed as $P\left(x_{i},y_{i},z_{\dot{l}}\right)$. mile can be represented as follows:(12)$$mile=\sum_{i=1}^{n}\sqrt{{\left(x_{\dot{i}+1}-x_{\dot{i}}\right)}^{2}+{\left(y_{\dot{i}+1}-y_{\dot{i}}\right)}^{2}+{\left(z_{\dot{i}+1}-z_{\dot{i}}\right)}^{2}}$$

## 3. Gray Wolf Algorithm

The advantage of wolf pack hunting is to rely on strong bullying; to attack less [27,28,29,30,31]; clear division of labor; mutual help; and group attacks. The success of a wolf pack lies in cooperation, which has two meanings: the first is the information sharing of a situation, knowing oneself and the enemy, and winning a hundred battles. The more abundant information, the lower the uncertain risk of hunting, and strategies to capture prey are much more feasible. The other is when multiple wolves attack their prey in the same place at the same time, and some are responsible for disrupting it [32]. Because a wolf’s hunting behavior depends on the autonomous cognition of the wolves, strict division of labor, and wolves on the whole have this distributed feature, wolves have internal collaborative features. This attracted the attention of the researchers, and inspired by the related researchers abstracted from wolves hunting wolves in the social hierarchy and wandering and calling, rounded up three kinds of intelligent behavior. These three kinds of behavior’s order cannot changed, and with strict dependence, are irreversible. When wolves wander, they find prey, summon them, and round them up [33,34]. Figure 2 shows the wolves’ hunting model.

### 3.1. Intelligent Behavior of Wolves

This part focuses on describing the three kinds of intelligent behaviors presented by wolf swarms with mathematical expressions, which lays a mathematical foundation.

**Wandering**. In the solution space, the best $S_{num}\;$wolf, except the head wolf, is regarded as the scout wolf; wandering is designed for attaining a more available solution. The position of the wolf after moving forward one step along the direction p(p=1,2,…,n) in the D-dimensional space can be expressed as follows [26]
(13)$$x_{id}^{p}=x_{id}+\sin\left(2\pi \times \frac{p}{h}\right)\times step_{s}^{d}$$
where $step_{s}$ is the walking step length and h is the number of directions.

**Summoning.** Summoning imitates the hunting that calls for companions to catch prey. The summoning is designed to renew the previous optimal value. For the k generation wolves, the position in the D-dimensional space can be expressed as follows [26]
(14)$$x_{jd}^{k+1}=x_{jd}^{k}+step_{b}^{d}\times \frac{g_{d}^{k}-x_{jd}^{k}}{\left|g_{d}^{k}-x_{jd}^{k}\right|}\;$$
where $step_{b}^{d}$ is the walking step length and $g_{d}^{k}\;$is the position of the k generation wolves in the D-dimensional space.

As wolves rush toward prey, they get closer and run quicker. To prevent missing prey, we set a determination distance. The determination distance $d_{near}\;$can be expressed as [26]
(15)$$d_{near}=\frac{1}{D\cdot \omega }\times \sum_{d=1}^{D}\left|M_{d}-m_{d}\right|$$
where D is dimension of the solution space; ω is the distance determination factor; $M_{d}$ is the maximum value; and$\;m_{d}$ is the minimum value in the d-dimensional.

**Rounding up.** The position of the wolf closest to the prey (the position of the lead wolf) can be regarded as the moving position of the prey. For k generation wolves, the prey’s position in the D-dimensional space is $G_{d}^{k}$, and then the wolf pack’s rounding up behavior can be expressed as follows [26]
(16)$$x_{id}^{k+1}=x_{id}^{k}+\lambda \times step_{w}^{d}\times \left|G_{d}^{k}-x_{id}^{k}\right|$$
where λ  is random number in the interval [1, 1] with uniform distribution, and $step_{w}^{d}\;$ denotes the walking step length.

In the d-dimensional space, the relationship among the wandering step $step_{s}^{d}$, summoning step $step_{b}^{d}$ and rounding up step $step_{w}^{d}$ of the wolf involved the three intelligent behaviors, which can be expressed as follows [26]
(17)$$step_{s}^{d}=\frac{step_{b}^{d}}{2}=2\times step_{w}^{d}=\frac{\left|M_{d}-m_{d}\right|}{C}\;$$
where C represents the *step* size factor, which represents the search refinement degree of the wolf in the solution space.

### 3.2. Adjust the Flight Time of Each UAV

The Gray Wolf algorithm can only obtain fitness and mileage information, and cannot directly access time information, or obtain speed, mileage information or the information time, but time is hard to agree upon, and needs to satisfy the speed constraints to adjust the speed, so that the times converge.

**Time adjustment factor**b**.**b is the time adjustment factor, which can adjust the velocity of the UAV to make the times converge. The *b* can be represented as follows
(18)$$b_{i}=\{\begin{array}{c} t(i)/t(j),t(i)<t(j)\\ 1,t(i)>t(j) \end{array}$$
when t(i)
*is* less than t(j), *b* can be represented as *t*(*i*)/*t*(*j*); otherwise *b* is one, where *i* represents the UAV’s serial number and *j* represents the neighbor of the UAV’s serial number.

**Update the velocity of UAV.** The UAV speed update depends on the time adjustment factor *b*, and the UAV time can be adjusted by adjusting the UAV speed. The new velocity of the UAV can be represented as follows
(19)$$new_{velocity_{i}}=\{\begin{array}{c} b_{i}*current\_velocity_{i},t(i)<t(j)\\ 1*current\_velocity,t(i)>t(j) \end{array}$$
when t(i) is less than t(j), $new\_velocity_{i}\;$ equal to $b_{i}*current\_velocity_{i}$, otherwise $new\_velocity_{i}$ equal to $1*current\_velocity_{i}$; where $new\_velocity_{i}$ represents the updated velocity of ${\mathrm{UAV}}_{i}$; and $current\_velocity_{i}$ represents the current velocity of ${\mathrm{UAV}}_{i}$. If the UAV completes the speed update, the UAV will then fly at the updated speed at a constant speed.

**Update the flight time of UAV.** The flying time of UAV is determined by both mileage and speed. The mileage of UAV is calculated by the Gray Wolf algorithm, and the speed of UAV is calculated by Formula (18). The update the flight time of UAV can be represented as follows
(20)$$new\_flight\_time_{i}=current\_mile_{i}/new\_velocity_{i}$$
where $new\_flight\_time_{i}$ represents the updated flight time and $current\_mile_{i}$ represents the mile optimized by the Gray Wolf algorithm.

### 3.3. Steps of GWO to Solve the Multi-UAV Path Planning Problem

The GWO algorithm is used to solve the multi-UAV cooperative path planning problem as follows.

**Step 1:** Set the start point and the goal point. Set the maximum iteration number of the algorithm $G_{\max}$, the total number of gray wolves N, the Wolf detection proportion factor α, the step size factor C, and the maximum walk limit number $T_{\max}$. Initialize the wolf pack population and number of UAVs.

**Step 2:** Judge whether $G\leq G_{\max}$ is satisfied. If it meets this rule, return to **Step 3**; otherwise, end the algorithm. The global optimal position obtained in **Step 7** is the global.

**Step 3:** Update the UAV number n=n+1 to judge whether it meets n < UAVs; if it meets this rule, go to **Step 4**; otherwise, go to **Step 5** and save the value of the mile.

**Step 4:** According to Equations (13)–(17), calculate the individual fitness of the wolves for the UAV itself, and update the mile and fitness; return to **Step 3**.

**Step 5:** Conduct interaction according to UAV interaction; according to the mile and the velocity of the UAV, calculate the flight time of the UAV.

**Step 6:** According to Equations (18)–(20), update the flight time of each UAV; update the iteration number of the algorithm G=G+1; return to **Step 2**.

**Step7: Output:** The flight time of each UAV, the mile of each UAV and the fitness of each UAV.

## 4. Simulation and Analysis

To evaluate the proposed method, a matlab-based simulator was built. The environment is an open space with the size of 100 M × 100 M × 100 M, which includes randomly generated obstacles. The simulation has a multi-agent system, which is composed of 4 UAVs. The start point is set up as [10,10,90,90; 10,90,10,90; 10,10,10,10]. The goal point is set up as [50,50,50,50; 50,50,50,50; 50,50,50,50]. Table 1 gives the GWO parameters. The parameters of UAV performance are shown in Table 2.

Figure 3a reveals a set of curves about the fitness change of UAVs based on GWO. Figure 3b is a set of curves about the fitness change of UAVs based on PSO. Figure 3c denotes a set of curves about the fitness change of UAVs based on AFO. Figure 3d shows a set of curves about the fitness change of UAVs based on GA. As can be seen from Figure 3, each algorithm can allow the fitness of the UAV to converge. Figure 4a reveals a set of curves about the flight time change of UAVs based on GWO. Figure 4b denotes a set of curves about the flight time change of UAVs based on PSO. Figure 4c is a set of curves about the flight time change of UAVs based on AFO. Figure 4d shows a set of curves about the flight time change of UAVs based on GA. As can be seen from Figure 4, GWO can make the UAV finally reach a consistent time. As can also be seen from Table 3, the variance of GWO is better than others. It can be seen from Figure 4b–d that PSO and AFO and GA do not completely match the final arrival time of UAV. Performance is not as good as GWO, so GWO is superior to PSO and AFO and GA. Figure 5 shows that the algorithm can plan the path of multiple UAVs, so that they can arrive at the same time.

Comparing the four algorithms, it can be found that PSO is easy to fall into the local optimal solution, while GWO, AFO and GA are easy to jump out of the local optimal solution. In order to make a more intuitive comparison, the GWO algorithm, PSO algorithm, AFO algorithm and GA algorithm were quantitatively analyzed, and the analysis results are presented in Table 3 and Figure 6. Figure 6a shows a set of box-plots about the value of the mean error of difference of the flight time between UVAs by GWO, PSO, AFO and GA. Figure 6b shows a set of box-plots about the difference of flight times between UAVs by GWO, PSO, AFO and GA. From the box plot, it can be analyzed that the median gap between GWO and AFO is very small during each generation iteration process, which is better than PSO and GA. The GWO is less volatile than the AFO, which means that the GWO is more stable. However, the outliers of GWO seem to be more than those of AFO, which is due to the insufficient convergence of GWO in the beginning. In order to better explain the problem, we need to explain the problem combined with Figure 4a. As can be seen from Figure 4a, the convergence of the GWO algorithm is insufficient in the beginning, and the convergence is fully achieved in the middle and late stages. This explains why there are many outliers, but the volatility is small by GWO algorithm. Although AFO performance looks like the GWO, combined with Figure 4b it will look to have a similar median and GWO, as AFO just makes part of the UAVs completely reach the time consistency, and make the average smaller, but the volatility will be very big, so the AFO box diagram box will look bigger than the GWO box, of which the AFO volatility is greater than the GWO. The closer the value of the mean error of flight time is to 0, the more time consistency requirements there are, as the value of the mean error of flight time should ideally be 0. It can be seen from Table 3 that the mean error of flight time between UAVs of the GWO is 0.213, which is superior to PSO (0.382), AFO (0.315) and GA (0.825). The simulation results are representative in the scenario of path planning with time windows for multiple UAVs. However, the current experiment number is still small, and the scene richness of the algorithm application is still insufficient. Further research should be made to reach a more general conclusion; within this range, GWO is superior to PSO, AFO and GA.

## 5. Conclusions

In this paper, we propose an optimization method to solve the path planning problem in complex environments. The Gray Wolf optimization (GWO) algorithm is employed to search the optimal path. In addition, we also introduce an interaction rule to exchange information to obtain the flight times of UAVs. Experimental results show that our method can effectively solve the problem of path planning with time windows in complex conditions, and the obtained simulation results are somewhat representative in the mountain scenario. The experimental results verify that the proposed method can plan a safe flight path in the process of multi-UAV flight, and reach the goal point in the same time. The mean error of flight times between UAVs of the GWO is 0.213, which is superior to PSO (0.382), AFO (0.315) and GA (0.825).

In future, research on path planning of multiple UAVs will use more advanced swarm intelligence algorithms to solve problems with more complex scenarios and more constraints, such as shallow water detection and forest rescue.

## Figures and Tables

**Figure 1 biomimetics-07-00225-f001:**
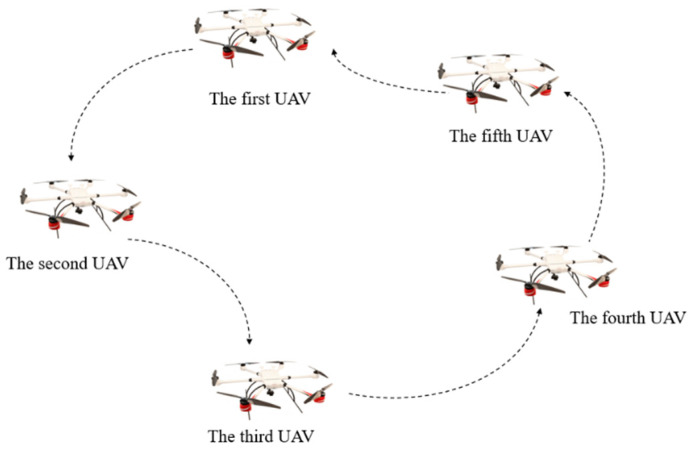
The network topology is the balance diagram.

**Figure 2 biomimetics-07-00225-f002:**
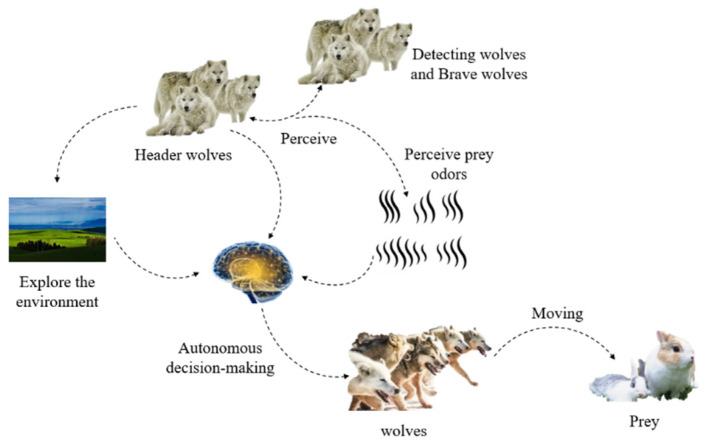
Wolves’ hunting model.

**Figure 3 biomimetics-07-00225-f003:**
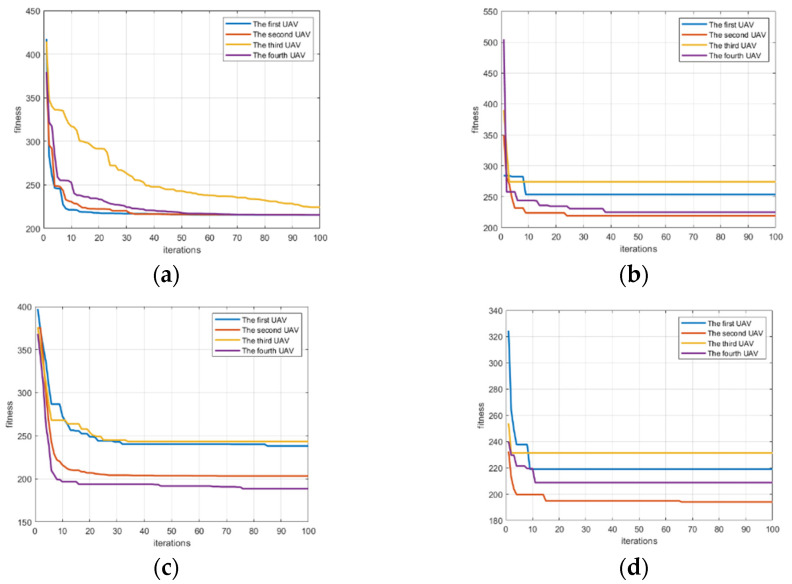
A set of change curves about the fitness of the UAVs by GWO, PSO, AFO and GA. (**a**) shows a set of change curve about the fitness of the UAVs by GWO; (**b**) reveals a set of change curve about the fitness of the UAVs by PSO; (**c**) indicates a set of change curve about the fitness of the UAVs by AFO; (**d**) reflects a set of change curve about the fitness of the UAVs by GA.

**Figure 4 biomimetics-07-00225-f004:**
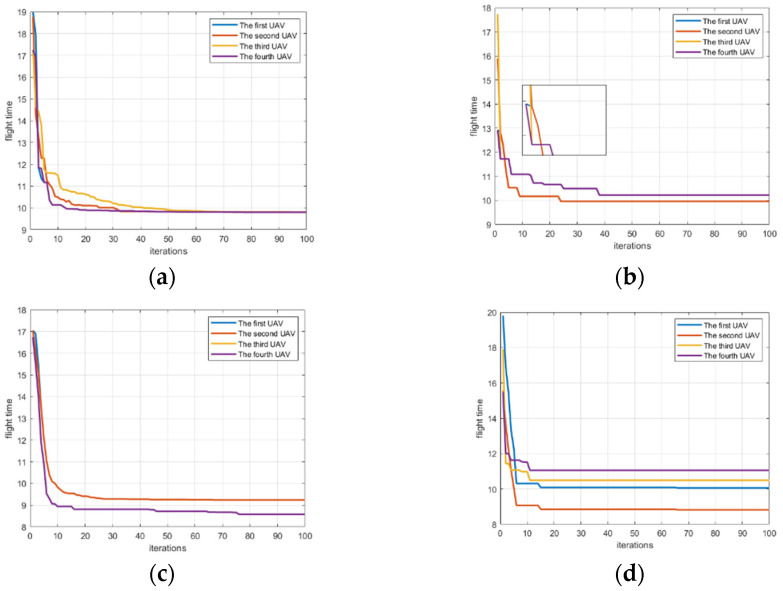
A set of change curves about the flight time of the UAVs by GWO, PSO, AFO and GA. (**a**) shows a set of change curve about the flight time of the UAVs by GWO; (**b**) reveals a set of change curve about the flight time of the UAVs by PSO; (**c**) indicates a set of change curve about the flight time of the UAVs by AFO; (**d**) reflects a set of change curve about the flight time of the UAVs by GA.

**Figure 5 biomimetics-07-00225-f005:**
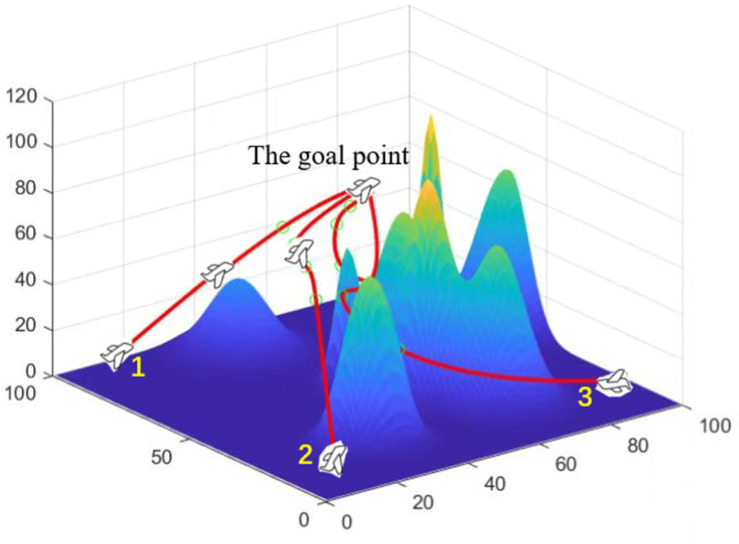
Time-consistent path planning for multiple UAVs.

**Figure 6 biomimetics-07-00225-f006:**
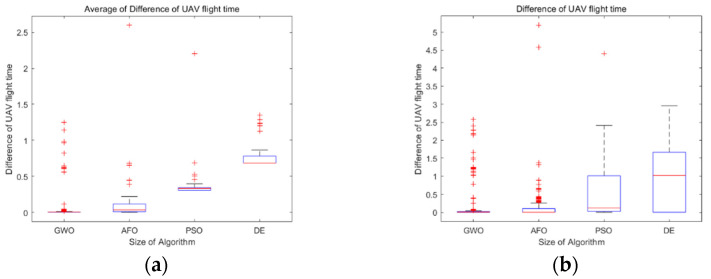
(**a**) shows a set of box-plots about average of difference of UAV flight times by GWO, PSO, AFO and GA. (**b**) shows a set of box-plots about difference of UAV flight times by GWO, PSO, AFO and GA.

**Table 1 biomimetics-07-00225-t001:** Parameters of GWO algorithm.

Parameters of GWO	Numerical Value
Iterations	100
Dimensions	3
Total number of wolves	100
Scale factor of detecting wolves	0.5
Step factor	20
Number of directions to summoning	20
Critical distance	10
The maximum number of rounding up	10

**Table 2 biomimetics-07-00225-t002:** Parameters of UAV performance.

Parameters of UAV Performance	Numerical Value
Minimum velocity	100
Maximum velocity	3
Maximum angle of turning	100
Maximum angle of downthrust	0.5
Minimum distance from the obstacle	20
Safe distance	20

**Table 3 biomimetics-07-00225-t003:** Simulation results comparison of difference of UAV flight times (400 times, all expected value is 0).

Method	Optimal Value	Mean Value	Worst Value	Variance
PSO	0.08	0.382	4.41	0.146
GWO	0.017	0.213	2.57	0.0453
AFO	0.05	0.315	5.19	0.0992
GA	0.029	0.825	2.69	0.703

## Data Availability

Not applicable.

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
