# Peer review of "Path Planning with Time Windows for Multiple UAVs Based on Gray Wolf Algorithm"

_biomimetics, 2022, doi:10.3390/biomimetics7040225_

Round 1

Reviewer 1 Report

I send the review in the attachment.

Reviewer 2 Report

The language needs extensive correction, long sentences make it difficult to understand the text. Style to be corrected in some places, e.g. there are no verbs or used twice. Repeating words occures frequently (for example: "to address this problem, the problem of ..." - line 10, "solved by solving" - line 37, "group behavior of the UAV group" - line 41). Invalid word forms (e.g. strength instead of strengthen). Formatting: e.g., case sensitive. Incomplete sentences (e.g. is defined as ... - line 90).

Notation of mathematical formulas (especially 18, 19, 7-9). Designations, some unexplained, some interchangeable upper and lower case letters, e.g. n and N.

The abstract and conclusions lack information on the results obtained (e.g. regarding the improvement of the results of times and the length of the UAV routes).

The word "wolf" is not necessary to be written with a capital letter when it refers to animals.

In computer simulations, I would consider different (random) locations of the drones' starting points, plus different obstacle positions. Such simulations would allow a better study of how the algorithm behaves, for example in relation to PSO.

No discussion with results of other authors (for example with other methods). Many algorithms are listed and comparison with only one PSO. There are no citations to some of the statements (eg line 38-39, 53, 62). Double numbering in Literature.

Please, see attachement for some detail information.

Round 2

Reviewer 2 Report

I still see a lot of shortcomings.

There is no information about the number of simulation repetitions and the use of statistical tests to show statistically significant differences in the obtained results.

4 different UAV starting positions were considered. In simulations, positions should be generated randomly, e.g. 1000 repetitions for different UAV locations and obstacles (not all of them have to be presented in the results). But on the basis of these average results from all variants, conclusions should be drawn about the superiority of the GWO method.

Sorry, but previously I had problems approving the review and the comment file was not attached. Hope this time it will be attached with no problem.

So, despite a lot of authors' work to improve the manuscript, I have reservations about the correctness of the inference.

Round 3

Reviewer 2 Report

Despite the work contributed by the authors to improve the article, I do not change my opinion regarding the correctness of the final conclusion. In my opinion, the authors performed too few computer simulations (20) to categorically conclude about the superiority of GWO over other algorithms. Perhaps it is so, but the presented analysis is only a small fragment (case study) of all possible cases to show that GWO is always better.

I presume that the authors will pursue further research in this direction, but in present form the manuscript has research deficiencies.

Also some questions have not been answered, such as: in formulas 7-9 I don't understand the notation - set intersection applied to numbers?

I leave the final decision of the article acceptance to the editors but I suggest rejection in current form.
